# CBCT for Diagnostics, Treatment Planning and Monitoring of Sinus Floor Elevation Procedures

**DOI:** 10.3390/diagnostics13101684

**Published:** 2023-05-10

**Authors:** Nermin Morgan, Jan Meeus, Sohaib Shujaat, Simone Cortellini, Michael M. Bornstein, Reinhilde Jacobs

**Affiliations:** 1OMFS IMPATH Research Group, Department of Imaging & Pathology, Faculty of Medicine, KU Leuven, 3000 Leuven, Belgium; 2Department of Oral Medicine, Faculty of Dentistry, Mansoura University, Mansoura 35516, Egypt; 3Department of Oral and Maxillofacial Surgery, University Hospitals Leuven, Campus Sint-Rafael, 3000 Leuven, Belgium; 4King Abdullah International Medical Research Center, Department of Maxillofacial Surgery and Diagnostic Sciences, College of Dentistry, King Saud bin Abdulaziz University for Health Sciences, Ministry of National Guard Health Affairs, Riyadh 11426, Saudi Arabia; 5Department of Oral Health Sciences, Section of Periodontology, KU Leuven, 3000 Leuven, Belgium; 6Department of Dentistry, University Hospitals Leuven, KU Leuven, 3000 Leuven, Belgium; 7Department of Oral Health & Medicine, University Center for Dental Medicine Basel UZB, University of Basel, 4058 Basel, Switzerland; 8Department of Dental Medicine, Karolinska Institute, 141 04 Huddinge, Sweden

**Keywords:** CBCT, three-dimensional imaging, maxillary sinus, sinus floor elevation, sinus floor augmentation

## Abstract

Sinus floor elevation (SFE) is a standard surgical technique used to compensate for alveolar bone resorption in the posterior maxilla. Such a surgical procedure requires radiographic imaging pre- and postoperatively for diagnosis, treatment planning, and outcome assessment. Cone beam computed tomography (CBCT) has become a well-established imaging modality in the dentomaxillofacial region. The following narrative review is aimed to provide clinicians with an overview of the role of three-dimensional (3D) CBCT imaging for diagnostics, treatment planning, and postoperative monitoring of SFE procedures. CBCT imaging prior to SFE provides surgeons with a more detailed view of the surgical site, allows for the detection of potential pathologies three-dimensionally, and helps to virtually plan the procedure more precisely while reducing patient morbidity. In addition, it serves as a useful follow-up tool for assessing sinus and bone graft changes. Meanwhile, using CBCT imaging has to be standardized and justified based on the recognized diagnostic imaging guidelines, taking into account both the technical and clinical considerations. Future studies are recommended to incorporate artificial intelligence-based solutions for automating and standardizing the diagnostic and decision-making process in the context of SFE procedures to further improve the standards of patient care.

## 1. Introduction

Alveolar bone atrophy in the maxillary posterior region is inevitable following tooth extraction, resulting in horizontal and vertical ridge resorption [1,2]. If a dental implant is inserted in bone having inadequate volume, there is a high risk of compromised implant stability and poor prognosis [3,4]. In order to compensate for reduced bone height and volume of the maxillary posterior region, sinus floor elevation (SFE) is performed for bone reconstruction. It involves lifting up of the Schneiderian membrane, which is usually followed by the placement of a bone graft [5,6,7,8,9].

The two main techniques for SFE are either a direct approach using a lateral window or indirect with the transalveolar technique. In the lateral window approach, an osteotomy is performed in the buccal wall of the maxilla, creating access through the lateral bone wall of the sinus cavity [10,11,12]. The transalveolar technique is a less invasive technique that was modified by Summers [13], where a transcrestal osteotome is applied to elevate the sinus floor, pushing bone substitutes beyond the level of the original sinus floor [14]. This technique has been recommended in areas with sufficient alveolar crest width and where a residual vertical bone height of ≥5 mm is available [15]. In cases where the alveolar bone height is less than 5 mm, the lateral window approach is recommended [16].

To date, various graft materials have been successfully used for SFE solely or in combination with each other [17], such as autograft [18,19,20,21] (intraoral: chin, retromolar region, mandibular ramus, maxillary tuberosity [22,23]; extraoral: iliac crest, fibula, tibia [24,25,26]), allograft [27,28,29] (fresh, frozen, freeze-dried bone [30,31]), xenograft [32,33,34,35,36] (deproteinized bovine bone [37,38,39]), and phytogenic material [39,40,41] (Gusuibu, coral-based bone substitutes, and marine algae).

Along with patient history and clinical examination, radiographic examination is an essential component of preliminary diagnostics, treatment planning, and outcome assessment in patients requiring SFE. Previously, two-dimensional (2D) panoramic radiography acted as a clinical standard. However, it suffers from certain inherent limitations, which can negatively impact the task at hand, such as magnification, distortion, and superimposition [16,42]. 

Three-dimensional (3D) preoperative imaging of the specific site of augmentation becomes a prerequisite in order to provide needed information about the morphologic characteristics and/or pathological conditions of the sinus and residual ridge [43,44,45]. Some studies have used computed tomography (CT) for the planning of sinus grafting [46] and precalculated the augmented bone volume needed [47,48,49,50]. However, the use of CT imaging is limited for most dental practitioners due to the high costs, large size of the device, and high radiation dose. To overcome these general limitations, 3D imaging in the form of cone beam CT (CBCT) has become a standard [43,44] for patients requiring maxillary sinus procedures. Recently, various CBCT imaging systems with low-dose protocols have also been made commercially available, which are not only affordable and compact to be used in a private dental practice but also provide 3D imaging with a lower radiation dose exposure to the patient [51,52].

The aim of the present review was to discuss the role of CBCT imaging for diagnostics, treatment planning, and postoperative monitoring of SFE as well as recommend future research directions, which could be useful for improving the current standard of patient care.

## 2. Use of CBCT for Diagnostics and Treatment Planning Prior to SFE

CBCT has been widely employed for diagnosis and preoperative treatment planning in all fields of dentistry [53], allowing for 3D visualization of dental structures and multiplanar reconstruction. It can provide cross-sectional images of the alveolar bone with the ability to accurately measure its height, width, and depict surrounding vital structures, such as the maxillary sinus. It allows for thorough assessment of the bone quality (density) and quantity (thickness) as well as the detection of bony changes, such as fractures or malformations. Furthermore, CBCT can be used for volume quantification that could help monitor bone remodeling and sinus disease [45,53,54].

Based on the recommendations of the American Academy of Oral and Maxillofacial Radiology (AAOMR) [55], CBCT imaging prior to SFE provides surgeons with a more detailed view of the surgical site. This helps to plan the surgical intervention more precisely while reducing patient morbidity. Additionally, it can be used to detect any potential problems and/or present pathology in advance, allowing for a safer and more successful bone grafting procedure. This also comes in accordance with the consideration of sinus augmentation mentioned in the updated guidelines for the use of diagnostic imaging in implant dentistry published by the European Association for Osseointegration (E.A.O.) [56].

From a technical point of view, in patients requiring CBCT acquisition for diagnostics, the “as low as reasonably achievable” (ALARA) principle has gone through a long history of evolution. Recently, some expert opinions advocated to rename it to “as low as diagnostically acceptable” (ALADA) [57] and, more recently, “as low as diagnostically acceptable being indication-oriented and patient-specific” (ALADAIP) [58]. These up to date principles highlight the need for actually optimizing rather than simply minimizing doses, taking into account not only the diagnosis but also treatment planning such as preoperative sinus grafting. For instance, when there is evidence of sinus pathology or sinus drainage is expected to be impaired, which might jeopardize the SFE outcomes, it is justifiable to extend the field of view (FOV) to include the whole of the maxillary sinus, including the ostio-meatal complex [59,60,61,62].

Furthermore, a multitude of CBCT devices exist in the market with variable scanning parameters, which include slice thickness, FOV, mAs, kVp, and scan time [63]. All these factors influence the image quality and amount of administered radiation dose. Generally, the effective radiation doses of CBCT devices for maxillofacial applications should preferably be 20 to 100 μSv; however, the doses of commercially available devices range from 10 to 1000 μSv depending on the parameter’s settings. Nevertheless, it is still lower than that of CT devices, which might vary between 474 and 1160 μSv [45,64]. Saying that, it is recommended to justify and optimize CBCT acquisition parameters for diagnostic tasks in an attempt to decrease the risk of high radiation dose exposure to the patient.

From a clinical point of view, CBCT enhances the diagnostic evaluation relative to 2D imaging by providing additional information related to the maxillary sinuses and surrounding structures. This diagnostic information could be a useful adjunct for SFE planning, as a thorough radiological assessment is not only important for the sinus surgery but also for implant placement. The diagnostic features extracted from CBCT images that could be clinically relevant for performing a successful SFE procedure, which could very well remain undetected with 2D imaging, are as follows:-*Anatomy of maxillary sinus and alveolar ridge*

CBCT imaging provides detailed anatomical information related to sinus anatomy. These findings allow the surgeon to assess the sinus morphology, density, and volume of the residual alveolar ridge, which might in turn help to determine the best approach for accessing the sinus and to evaluate the suitability of the patient’s bone for grafting [55,65].

-
*Relation of maxillary sinus to the roots of adjacent teeth*


If there is an intimate contact between the root(s) of the teeth and the Schneiderian membrane, the risk of membrane perforation during the sinus lift procedure is increased. Hence, it is important to evaluate the root proximity to the sinus during the diagnostic phase through CBCT imaging for decreasing the risk of perforation [66,67]. Additionally, the health state of adjacent teeth should be examined for the presence of pre-existing apical pathology that could result in sinus graft infection [68].

-
*Thickness of the Schneiderian membrane*


CBCT has been reported to be a useful tool for assessing the thickness of the Schneiderian membrane [60,69], which has also been reported to be associated with the occurrence of membrane perforation [70]. Healthy sinus mucosa has a mean thickness of 1 mm, although there is a wide range of variability among individuals [71]. Meanwhile, it should be noted that the risk of sinusitis following SFE increases when the membrane thickness is over 2 mm [72], and a higher risk of ostium obstruction exists if the membrane thickness is over 5 mm [73].

-*Maxillary sinus septum* [74,75]

Prior evidence suggests that at least one-third of patients have sinus septa, which are visualized ideally through 3D imaging [76]. The knowledge about septa location and morphology is a key factor in planning SFE, as it is associated with an increased risk of sinus membrane perforation during the procedure. Furthermore, if present, the osteotomy design might also require an alteration from a single window technique to two smaller windows on either side of the maxillary sinus floor septum or the use of a W-shaped trapdoor technique. Hence, CBCT imaging is the key to success in devising a proper treatment plan [77].

-
*Maxillary sinus ostium*


Sinus healing following SFE is largely dependent on sufficient drainage of the nasal cavity. If the ostium is not patent, the drainage would be impaired, which could cause sinusitis or surgical failure [72]. For guaranteeing appropriate mucociliary drainage and clearance, the ostium patency must be evaluated prior to surgery. In addition, the sinus should be assessed for the presence of accessory ostia, which can interfere with sinus ventilation and drainage [78].

-
*Maxillary sinus floor width*


The distance and angulation between the lateral and medial maxillary sinus walls are also important anatomical features to be evaluated with CBCT imaging. It allows determining the difficulty level of performing SFE, as too narrow or too large sinuses with sharp angulations are considered complex cases. Moreover, accurate measurement of the sinus width based on CBCT imaging is also crucial for deciding the surgical approach; for example, a trapdoor SFE technique is contraindicated in patients having narrow sinuses [79].

-
*Thickness of the lateral maxillary sinus wall*


Maxillary sinus lateral wall thickness is an important parameter to be assessed using CBCT imaging at the diagnostic stage because SFE through a thick wall is difficult to perform, takes longer time and is more prone to perforation. Hence, CBCT imaging is suggested to help with the decision-making process, as it allows the surgeon to 3D evaluate the sinus wall and select the region with the least thickness to avoid complications [80].

-
*Alveolar antral artery*


CBCT imaging allows a clear depiction of the antral artery, allowing for an optimal planning of the surgical access to the sinus. Alveolar antral arteries with a diameter more than 0.5 mm can be observed on CBCT, and profuse bleeding should be expected if the artery has a diameter more than 3 mm [81]. If present at the osteotomy site, the use of a piezosurgery device is preferred. Moreover, changing the osteotomy window design from an oval to a round shape through either above or below this artery could avoid injury [82].

-
*Estimation of graft volume*


Using the combination of CBCT images along with the various planning software systems available allows for measuring and extracting the sinus volume necessary to be grafted [83]. Adequate preoperative planning of the graft volume may help to avoid sinus over-filling and potentially occluding the ostium, decide on the ratio of bone and bone substitutes to be mixed, and estimate the cost of xenografts prior to the actual operation [47,48]. It is worth noting that in cases where an autogenous graft will be harvested, preoperative knowledge of the amount of graft required is useful in selecting the optimal donor region, reducing the time and complexity of the surgical procedures, as well as minimizing potential postoperative complications [84]. 

-
*Incidental findings*


The role of CBCT in revealing incidental findings (IFs) which are not related to the primary scan indication also cannot be ignored. Several studies have reported a high prevalence of IFs in the maxillary sinus region on CBCT images when acquiring the scan for the purpose of implant/surgical planning [85,86]. The most common IFs encompass concha bullosa, mucosal thickening, polyps, altered sinus dimensions, and sinus opacification. Some IFs may also be suggestive of benign or malignant neoplastic processes. Hence, it is important for the dental practitioner to be aware of these findings on CBCT images, which might allow a more appropriate selection of a treatment plan as patients with IFs might be redisposed to a higher risk of postoperative complications from SFE and implant placement. Although a smaller FOV CBCT scan results in less radiation, the risk of missing IFs still exists. Hence, it is recommended to acquire a scan with an optimal FOV depending on the clinical indications and risk–benefit analysis. To reach a more definitive conclusion, more research is also needed to examine how different FOVs affect the incidence of IFs.

## 3. Digital Workflow for SFE Procedures

In recent years, digital technologies and workflows have been introduced in the majority of dental medicine fields, including restorative dentistry, orthodontics, dental implantology, and maxillofacial reconstructive surgery [87]. Digital treatment planning workflow refers to the incorporation of computer-controlled components and dental technologies for assisting a clinician with the planning process. This digitization of workflows in clinical dentistry has overcome the limitations associated with traditional methods by offering improved precision of dental procedures, time-efficiency, and a higher standard of patient care [88,89,90].

### 3.1. Virtual Modelling

Generally, CBCT images are saved in a Digital Imaging and Communications in Medicine (DICOM) format, which is then transferred to 3D software programs for further processing to plan the procedure. The most essential step in SFE planning workflows is segmentation, a process by which the region of interest is extracted from 3D images for generating 3D virtual models. These models are then used for fabricating guides or pre-surgically assessing the amount of required (bone) graft (Figure 1).

Traditionally, manual sinus segmentation, referred to as slice-by-slice delineation on 2D CBCT planes, performed by an expert is considered the gold standard. However, it is prone to certain limitations, such as labor-intensiveness, increased time consumption, and observer variability [91,92]. Based on the aforementioned limitations, semi-automated segmentation via thresholding-based approaches has been widely adopted for the segmentation of CBCT images to improve the efficiency of planning workflows [93,94]. Nevertheless, the final segmentation lacks optimal delineation due to the presence of different structural densities, and manual post-processing is often required. Recently, artificial intelligence (AI) in the form of deep learning has been employed for automated segmentation to overcome the limitations associated with both manual and semi-automated segmentation approaches. In deep learning, convolutional neural networks (CNNs) have demonstrated excellent performance with the employment of multi-layer neural computational connections for sinus segmentation on CBCT images [95,96,97]. The application of such deep learning-based approaches might enhance the quality and predictability of presurgical graft planning, enable a more precise treatment planning process and volumetric quantification of sinus/graft changes, and may further improve the standard of care. Yet, a lack of evidence exists related to the application of AI in the SFE treatment planning workflows.

### 3.2. Surgical Guidance

In SFE procedures, CBCT-based guidance has played a vital role in improving the precision of the surgical procedure with a reduction in complications. The guidance can be static or dynamic in nature. The procedures performed via these guides are referred to as “guided sinus lift [98]” or “guided bone grafting [99]”.

#### 3.2.1. Static Surgical Guides

Static guides are designed via 3D planning software programs following the integration of intraoral scanned images with CBCT datasets and later fabricated using 3D printers. Such surgical templates act as a support aid and offer the advantages of time-efficiency, better working ergonomics, less operator stress, and greater predictability of the procedure [98,100,101]. Moreover, this procedure can also be combined with concurrent implant placement planning.

In 2008, Manderales and Rosenfeld [99] pioneered computer-guided SFE. They proposed using CAD/CAM surgical cutting guides for exact lateral wall outlining to considerably improve the quality and outcomes of the SFE procedure. Cecchetti et al. [98] have recently introduced virtual planning of surgical guides for lateral wall sinus elevation, concluding that the surgical template should be seen as a support aid to minimize risk and complications of the surgical procedures, especially in “difficult” cases. 

Following the same concept, Osman et al. [102] and Strbac et al. [101] performed computer-guided SFE through a lateral window approach in addition to simultaneous implant placement. They found that applying static guides resulted in better and more consistent results. Similarly, Pistilli et al. [100] also concluded that such a digital approach is highly efficient in the mid-term (follow-up of 10 years) to implant rehabilitation of severely resorbed maxilla simultaneously with sinus lift. 

Considering transrectal sinus augmentation, Pozzi et al. [103] and An et al. [104] combined static guide-based flapless maxillary crestal sinus augmentation with an immediate nonfunctional loading of dental implants, reporting a 98.53% and 100% survival rate at 3 years and 37 months, respectively. 

Another type of surgical guidance is dynamic navigation, which is based on computer-guided surgery planning. Here, a physical surgical guide is unnecessary [105].

#### 3.2.2. Dynamic Surgical Guides

A dynamic navigation system combined with CBCT imaging has been proposed for improving the intraoperative precision of implant placement, where a static surgical guide is not required and the operator can place the implants with real-time navigation [106]. With regard to SFE, limited evidence exists related to the application of navigation-based approaches. Recently, dynamic navigation has been used for posterior maxilla implant surgery via transcrestal SFE using piezoelectric devices [107]. The proposed technique offered high precision with excellent clinical outcome. However, it is recommended to perform further clinical studies to assess the effectiveness of dynamic navigation for performing SFE. 

Considering the intraoperative use of CBCT imaging, Blake et al. [108] performed one case trial of using a C-arm-based CBCT scanner during sinus augmentation surgery under general anesthesia with iliac crest grafting. The images were taken prior to wound closure to immediately verify the surgery result. However, there is no solid evidence regarding such procedures for surgeries performed under local anesthesia.

## 4. Use of CBCT for Monitoring and Follow-Up after SFE

Post-surgical radiographic examination is vital [19,49,109,110,111,112,113,114,115,116,117,118,119,120,121] for the evaluation of bony integration of the inserted graft, follow-up of its long term stability, and also assessment of implant success and/or osteointegration after sinus lift procedures. 

Ideally, imaging guidelines for the follow-up of sinus augmentation with or without immediate implant placement should follow the same regulations as those for post-surgical implant placement. Based on the AAOMR recommendations [55] and the guidelines for the use of diagnostic imaging in implant dentistry published by the E.A.O. [56], intraoral periapical radiography should be performed for the postoperative assessment of implants in the absence of clinical signs or symptoms. Panoramic radiographs may be indicated for more extensive implant therapy cases.

Meanwhile, CBCT [109,112,115] has become the standard imaging technique for 3D visualization and improved assessment prior to implant placement in the case of staged sinus augmentation. CBCT imaging can help to assess bone healing by visualizing how the material has integrated with the surrounding bone as well as if any signs of early resorption exist. It is beneficial in providing information about the volume, extent, and density of the augmented region [56] and can also be used to monitor any complications, which are not visible to the naked eye, such as mucosal changes, infection, and/or inflammation. 

Furthermore, bone graft materials undergo remodeling over time at varying rates depending on the material used (resorbable versus non-resorbable) [122], and positive pressure formed inside the sinus during respiration also accelerates graft resorption [123]. Certainly, this remodeling could have a significant impact on the success of SFE outcome and the respective implant treatment. Thereby, CBCT can aid in the quantification of the resorption rate of different grafting materials [124,125,126] and also to monitor sinus changes at follow-up stages by comparing pre-surgical and/or post-surgical scans acquired at different time points. Usually, the Schneiderian membrane exhibits significant post-surgical edema, which increases the mucosal thickness visible. In addition, the edema might cause ostium obstruction with the possibility of impaired drainage capacity of sinus mucus. The reduction in the patency and obstruction of the ostium and infundibulum can lead to an inflammatory reaction and/or infectious processes of the sinus cavity, causing acute or chronic sinusitis [127]. Hence, CBCT imaging could act as a useful follow-up assessment tool to measure the thickness of the membrane and monitor mucosal changes in an attempt to avoid both early and delayed postoperative complications [128]. Additionally, following up the mucosal thickness changes could help to study the possible effects of different graft materials on sinus mucosa for research purposes [129].

It should be kept in mind that the use of CBCT should not be opted for regular follow-up assessment of normal SFE procedures without any evident or suspected complications to avoid exposure to unnecessary radiation doses. Postoperative CBCT could be indicated in cases with complications and contraindicated in patients where no direct benefit is to be expected. In instances of no clinical signs or symptoms of treatment failure or complications, periapical or panoramic images could be considered more than enough for postoperative follow-up. Moreover, CBCT imaging should also be justifiable for ethically approved clinical research projects, which might improve the standard of patient care. However, optimized strategies still need to be developed for SFE follow-up assessment, allowing good image quality and accurate 3D modeling with low-dose scanning protocols. 

## 5. Recommendations and Future Developments

Maxillary sinus floor elevation is nowadays considered a safe and effective surgical technique to allow prosthetic restoration supported by implants in the atrophic posterior region of the maxilla. CBCT imaging has significantly improved the accuracy and efficiency of performing SFE procedures. It has become an imaging modality of choice for diagnostics due to its potential for detecting challenging anatomical and/or pathological entities, which would probably remain undetected with 2D imaging. However, one should consider that in a private practice, it is difficult and time consuming for a dentist to identify all CBCT-based diagnostic parameters, which might negatively impact the decision-making process. Hence, future studies should attempt to integrate artificial intelligence-based solutions using CBCT images for automating and standardizing the diagnostic and decision-making process for performing SFE procedure.

In relation to treatment planning, automated AI-based CBCT image segmentation has already been implemented for the production of virtual models of maxillary sinuses and jawbone. However, still no evidence exists related to the integration of AI networks in the digital treatment planning workflows of SFE. Hence, future studies should also focus on investigating the accuracy and efficiency of these networks and models in treatment planning workflows. Furthermore, they should also investigate the combination of quantitative extracted features following image segmentation with biological, clinical, and demographic patient characteristics, which is referred to as ‘’radiomics’’ [130]. Such a combination could be a step forward toward personalized dental medicine, enabling the best possible treatment for each patient [131,132].

As for follow-up assessments, it is recommended to perform studies comparing the 3D resorption rate associated with different surgical techniques and grafting materials and also assess the morphological changes of the sinus. Even for the post-surgical assessment of the sinus and grafted region, the application of AI allowing automated virtual modeling could be beneficial in an attempt to further improve the surgical outcomes and better understand the associated complications. As for the justification of CBCT imaging, there is a need for future research to establish an optimized low-dose CBCT protocol for SFE diagnostics, 3D planning, and follow-up assessment, which does not impair the image quality depending on the task at hand.

## Figures and Tables

**Figure 1 diagnostics-13-01684-f001:**
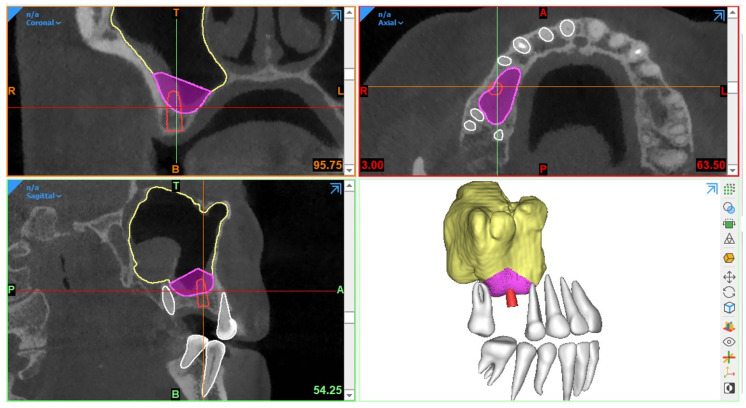
Example of graft volume estimation in Mimics (version 23.0, Materialise N.V., Leuven, Belgium) following automated sinus and teeth segmentation (creator.relu.eu, Relu, BV, Version March 2023).

## Data Availability

Data sharing is not applicable to this article.

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
