# Peer review of "CBCT for Diagnostics, Treatment Planning and Monitoring of Sinus Floor Elevation Procedures"

_diagnostics, 2023, doi:10.3390/diagnostics13101684_

Round 1

Reviewer 1 Report

Line 75: CT is mentioned but one aspect that is missing is that it is performed by radiologists, not by dental surgeons. This has the advantage to keep its use limited. Small devices whose use can be indicated by the surgeon will inevitably yield more radiation due to financial interest. The following lines which describe possible widening of indication are proof for this approach. General advantages of CBCT regarding radiation are thus used up rather easily. 

line 103 104: ALADA is not widely known, Wikipedia for example knows only the ALARA principle which covers more ground. ALDAIP - same.

The question of incidental findings has to be discussed, another matter radiologists are familiar with; to dental surgeon this certainly is unfamiliar ground. This holds true especially if the anatomy being imaged is increased. There are no numbers given for radiation doses CT/CBCT- please add.

Author Response

Thanks so much for your comments. 

Line 75: We appreciate the reviewer comments. The focus here is imaging for dental practitioners, at this moment CBCT has still lower radiation dose, is more compact and less costly, thus better serving a private practice. Besides, the footprint of devices also technical support / maintenance needed are relevant factors in practice, and thus, have given CBCT a great advantage in comparison to CT with regard to availability in dental practices / hospitals / universities. We adjusted the statement to be more clear.

Line 103 104: As suggested, we added ALARA concept and explained more about the difference with recent concepts.

As suggested, we added a paragraph about incidental findings and a paragraph about the doses. 

Reviewer 2 Report

This study consists in a description work that brings together a series of topics that are well known in the literature

It is well described and also exhaustive.
A good descriptive work of the subject treated with a very accurate bibliography.
In literature there are several descriptive works of the article however the description of the subject is done quite well. While not adding anything new from the point of view of research, however it can be considered a fairly exhaustive article in the description of the method

Author Response

Thanks a lot for your comment. 

Reviewer 3 Report

Dear authors,

It was a pleasure to read the review. The article demonstrates the important roles of the CBCT in SFE procedures.

The introduction is very well written, providing informations about SFE, grafting materials and the role of the CBCT in sinus augmentation. 

The chapter 2 is very well structured, bringing a few aspects that must be considered in planning of the SFE procedures: the anatomy of the maxillary sinus and alveolar ridge, the relation of the maxillary sinus and the roots of the molars, the thickness of the sinus mucosa, the septum, the ostium and the alveolar antral artery. This elements are key points with a wide variability and can alter the success of the surgical procedure.

The next chapter is focused on the digital workflow. It is an important part of the review taking into consideration the spread of the digital technologies.

From my point of view, the article is suitable for publication in the present form, being very well structured and written in a clearly manner and english.

Author Response

Thanks so much for your comments, really appreciated. 

Reviewer 4 Report

The study deals with an important procedure in oral surgery and a frequently used imaging tool. The work is well written. And could actually be accepted in this form. However, I would suggest slightly expanding the scope of the manuscript to include radiomics [there is only 1 article cited] and trying to describe methods of standardization of CBCT examination. Are there any known methods for geometric, exposure or optical density standardization in the volume of interest?

Author Response

We appreciate the reviewer comments.

As suggested, we mentioned radiomics and added further references. 

Considering standardization of CBCT examination, we have mentioned the technical/clinical aspects to be considered. We recommended to justify and optimize CBCT acquisition parameters for both diagnosis and also treatment.